# Managers’ Action-Guiding Mental Models towards Mental Health-Related Organizational Interventions—A Systematic Review of Qualitative Studies

**DOI:** 10.3390/ijerph191912610

**Published:** 2022-10-02

**Authors:** Melanie Genrich, Peter Angerer, Britta Worringer, Harald Gündel, Friedrich Kröner, Andreas Müller

**Affiliations:** 1Institute of Psychology, Work & Organizational Psychology, University of Duisburg-Essen, 45141 Essen, Germany; 2Institute for Occupational, Social & Environmental Medicine, Centre for Health and Society, Medical Faculty, Düsseldorf University, 40225 Düsseldorf, Germany; 3Department of Psychosomatic Medicine and Psychotherapy, Ulm University Medical Center, 89081 Ulm, Germany

**Keywords:** work design, leadership, health promotion, attitude, implementation, social norms, behavioral control

## Abstract

Research indicates that managers’ active support is essential for the successful implementation of mental health-related organizational interventions. However, there is currently little insight into what subjective beliefs and perceptions (=mental models) make leaders support such interventions. To our knowledge, this is the first qualitative systematic review of this specific topic, and it considers 17 qualitative studies of managers’ perspective. Based on the theory of planned behavior, this review provides an overview of three action-guiding factors (attitudes, organizational norms and behavioral control) that can serve as starting points for engaging managers in the implementation of mental health-related measures and ensuring their success. Our results provide evidence that supportive organizational norms may particularly help to create a common sense of responsibility among managers and foster their perceived controllability with respect to changing working conditions. Our study thus contributes to a more differentiated understanding of managers’ mental models of health-related organizational interventions.

## 1. Introduction

The ongoing fundamental transformation of work, which is characterized by intensified global competition, the rising importance of the service sector, and a fast-changing working environment due to newly digitalized workflows, has gone hand in hand with a substantial increase in psychosocial work stressors [1,2]. There is robust empirical evidence that such psychosocial risk factors are associated with individual employees’ stress experiences and in the long run might contribute to even more serious consequences, such as burnout or mental disorders [3]. Therefore, organizations across all sectors need efficient occupational safety and health (OSH) measures to maintain or improve their employees’ mental health and thus increase productivity.

The vast majority of scholars and practitioners agree that OSH measures should primarily seek to improve working conditions, which can eliminate structural risk factors for work stress and therefore sustainably improve employees’ mental health [4,5]. Such OSH measures focusing on improving working conditions, such as work tasks, structures and processes, to maintain or improve employees’ mental health are often referred to as mental health-related organizational interventions [6,7,8,9]. However, despite their importance, empirical evidence on the effectiveness of organizational mental health-related interventions is mixed at best [10]. This is likely due to an insufficient understanding of how to successfully implement such interventions [4,11,12]. 

There is a broad consensus that managers are one of the main drivers of the successful implementation of mental health-related organizational interventions, and organizational change interventions in general [4,13,14]. At best, managers transparently inform employees about opportunities and risks associated with the intervention and motivate them to participate in developing solutions to make working conditions “healthier”, e.g., by removing obstacles in work processes, such as insufficient information [7,15,16]. Managers provide the time and resources enabling participation and decide whether the developed work design solutions will be implemented [17,18,19]; they also help to integrate interventions’ project structures (e.g., steering groups, workshops) into existing organizational and managerial structures in order to create synergies and to ensure smooth intervention processes [18,20,21]. 

Existing organizational research demonstrates that managers’ mental models of an intervention are decisive for ensuring their active support [4,22]. A definition of an individual mental model is a concentrated, personally constructed, internal conception of external phenomena or experiences (past, present or projected) that affects how a person acts [23]. This definition is rooted in action and motivational theories, a core assumption of which is that people’s intentions to perform an action depend on their situation-specific cognitions regarding expected outcomes, opportunities to act, and conducive or obstructive contextual conditions [24,25,26]. For example, if a manager expects that an intervention will lead to a meaningful outcome, he or she may be motivated to actively support the intervention [27]. 

Currently, there is a lack of systematic understanding as to which specific aspects of mental models are linked to managers’ actual behavior in the context of organizational interventions [4,22], although studies have pointed to the relevance of attitudes and cognitions in the form of readiness or resistance to change [28,29]. There is extensive literature examining leaders’ roles, responsibilities and characteristics in relation to change processes. However, this research is primarily descriptive and based on observations of managers (e.g., their knowledge, skills, abilities and effectiveness) [30]. Thus, we do not know which thoughts are decisive for managers’ active support for organizational interventions or in their refusal of support or even active hindrance of an intervention. This is particularly true for organizational mental health-related interventions, which we argue have specific characteristics and therefore require special consideration in light of the situational specificity of action-guiding cognitions: As improving employees’ mental health usually does not relate directly to an organization’s core goals, the involved manager often experiences role conflicts, has inadequate skills and needs to establish new project structures that go beyond daily business processes [27]. Moreover, designing healthier working conditions—such as reducing time pressure—is often perceived as particularly difficult and complex to implement, expensive or time-intensive [27,31]. Thus, participating in mental health-related organizational interventions often involves specific challenges and burdens for managers. Against this background, our study aims to provide a systematic understanding of action-guiding aspects of managers’ mental models regarding mental health-related organizational interventions. 

In order to do so, we integrate the theoretical perspective of the theory of planned behavior (TPB) [26] into the organizational literature on mental health-related organizational interventions. The TPB is one of the most extensively studied models of human behavior and has been shown to explain and predict health-related behavior in a wide range of domains such as physical exercise, as well as technology adoption [26,32,33,34]. In a nutshell, the TPB suggests that human behavior is basically guided by three kinds of cognitions: personal attitudes towards a behavior, the perceived *normative beliefs* of a referent group towards the behavior, and the perceived control in successfully carrying out the behavior (based on internal resources such as skills or external resources such as time) [26,34]. So far, the theory has been applied only a few times to explain behavior in the context of organizational change, such as the implementation of organizational interventions [35]. In a rare example, Jimmieson et al. (2008) show from an applied perspective that the TPB can be a useful framework for pre-implementation assessments of employees’ readiness for change. We therefore argue that the TPB perspective can provide new insights that contribute to a better understanding of managers’ behavior in implementing mental health-related organizational interventions and organizational interventions in general. Due to its high level of empirical support in a variety of domains, the TPB helps us cluster the multitude of components involved in managers’ mental models in a meaningful and theoretically sound way. This allows us to interpret managers’ thoughts and cognitions with respect to their relevance for the active support of organizational health-related organizational interventions.

Our study can therefore contribute to research on occupational health and organizational change interventions in the following ways: With respect to occupational health research, our study can inform the development of instruments to measure managers’ readiness to support mental health-related organizational interventions. Moreover, differentiated knowledge of managers’ mental models can inform the development of preparatory sensitization or training interventions to meet managers where they are and engage and empower them to implement interventions that promote mental health. This can help prepare the ground for the more implementation of mental health-related organizational interventions in practice, which is urgently needed at present. With respect to change management research, using TPB our study can provide support that managers’ readiness for change acts as an antecedent for intention to support change. Identifying the beliefs’ underlying attitudes, subjective norms and perceived behavioral control can help to develop a better understanding of the psychological factors that distinguish managers who support change from those who do not. Such assessments could help change leaders to select strategies and tactics to engage managers, as key change agents. This subjective perception complements the previously dominant descriptive perspective in the change management literature and contributes to the theoretically informed development of models for successful change processes. 

In the following sections, we will first describe the specific characteristics of mental health-related organizational interventions compared to organizational change interventions in general. Then, drawing upon TPB, we discuss in more detail the role of managers’ mental models for the implementation of mental health-related organizational interventions. After that, we will present the results of a systematic review of qualitative studies on managers’ perceptions of mental health-related organizational interventions. We consider qualitative studies to be a valuable complement to the mostly quantitative studies of stakeholder reactions and behaviors regarding general organizational change processes that exist today (e.g., Oreg et al. (2011), or change management studies examining which leadership behaviors or other “change drivers” make change processes successful [14,30]. We will deductively analyze the content of the identified studies from the perspective of TPB in order to systematically identify relevant action-guiding aspects of managers’ mental models. Moreover, we will identify further aspects of these mental models that are not covered by the TPB in an additional inductive analysis. 

### 1.1. Mental Health-Related Organizational Interventions—What They Are and What Makes Them Unique?

Mental health-related organizational interventions are based on established models of job stress, work motivation and action regulation [5,36,37,38]. Accordingly, numerous empirical studies have shown that psychosocial job characteristics such as high workload, time pressure, high work demands with low control, work interruptions, mismatch between effort and reward, insufficient social support or poor management style can impair the psychosocial health of employees [36,39,40,41,42]. Basically, mental health-related organizational interventions aim to identify critical working conditions in organizations and modify them to promote employees’ mental health or reduce mental health risks. Policymakers and OSH experts often point out that mental health-related organizational interventions are preferable to individual interventions that promote employees’ skills in coping with work stress, e.g., stress management or resilience training [5]. This is because a larger number of employees benefit from mental health-related organizational interventions and because the effects should be more sustainable, as structural causes of stress and mental health impairment at work are mitigated [43]. According to a systematic review by LaMontagne, Keegle, Louie, Ostry, & Landsbergis (2007), when implemented optimally, mental health-related organizational interventions exhibit positive effects at both the organizational and individual level, whereas individual-focused interventions only improve individual persons’ abilities, with the external causes of stress in the organization remaining present. However, when mental health-related organizational interventions are poorly implemented, no or undesirable effects are often reported [20,44]. 

### 1.2. Mental Health-Related Organizational Interventions—What Is It about Implementation?

Like most organizational interventions, mental health-related organizational interventions intervene in the complex existing social structure of organizations and are in turn influenced by that structure [45]. Previous research suggests that due to this complexity, both company stakeholders (e.g., managers) and researchers prefer individual interventions because they are easier to implement in organizations [17,46,47]. Various authors therefore emphasize the similarities between mental health-related organizational interventions with common organizational change management approaches. Such similarities include the strategic priority of the change, commonly shared goals, commitment of key stakeholders to participate, a transparent communication structure, and the provision of adequate personnel, financial, and temporal resources [4,48].

However, we agree with Montano (2018) that although organizational change research has already provided comprehensive and systematic insights into the design, steps and drivers of organizational change processes in general [14], these insights cannot be directly applied to mental health-related organizational interventions: One key success factor for organizational interventions in general are affected stakeholders’ attitudes or other cognitions towards the intervention [4,22]. If stakeholders’ attitudes or cognitions are negative, there will be no support, resistance or negative side effects may arise and the intervention may fail [14,22]. Attitudes, in turn, seem to depend on the specific content of the intervention. For example, if an intervention’s goal or content is not perceived as meaningful, this can lead to negative reactions [22]. The specific goal of mental health promotion is usually not the most important organizational goal or may even be perceived as conflicting with primary organizational goals such as increasing sales [27]. Additionally, designing healthier working conditions is often perceived as particularly difficult and complex to implement, characterized by a lack of financial and time resources [27,31]. For this reason, it can be assumed that mental health-related organizational interventions pose special challenges for managers and therefore require special consideration.

### 1.3. Implementation of Mental Health-Related Organizational Interventions—Why Focus on Managers’ Mental Models?

Current health-related implementation research shows that managerial engagement is of utmost importance for the success or failure of health-related organizational interventions [7,13,15,16,49]. These findings are consistent with the extensive research on change management, which identifies specific managerial tasks and behaviors as predictors of effective organizational change implementation [14,30]. While managerial behavior has been frequently and extensively investigated in the literature as a driver of change processes, few studies have addressed the predictors of such supportive behavior [4,22]. For example, a review by Oreg et al. (2011) examined antecedents of different stakeholders’ reactions to general organizational changes. The authors suggest that stakeholders’ affective, cognitive and behavioral reactions, in turn, trigger their actual behaviors influencing the change process. Stakeholders here include all employees involved in the change process, and the review did not focus on or specifically examine managers, despite the fact that they play a dominant role in change processes. Accordingly, Montano (2018) proposed a multi-level model of the main drivers of organizational change, encompassing characteristics of the organization, the group and the individual that have been associated with attitudinal and behavioral change in empirical research [4,50]. The model notes the importance of (shared) mental models by managers for achieving the change goal or preventing conflicts during the change process. However, the specific components of the mental models and how these components are linked with behavior are not discussed or specified in detail.

The theory of planned behavior (TPB) [26] can be used to systematize and explain managers’ mental models that lead to health-promoting behavior and support for health-related organizational interventions. The TPB assumes that attitudes, subjective norms and perceived behavioral control (PBC) are predictors of behavioral intentions. Intention then acts as a mediator between attitudes, subjective norms, and PBC on the one hand and the actual execution of behavior on the other. The TPB model indicates that any behavior can be predicted using the three main components of attitudes, subjective norms and PBC. An attitude is a person’s belief that a behavior leads to a certain outcome, which is assessed as positive or negative. Subjective norms are “perceived social pressure to deal with behavior or not” [51]. PBC is the experienced ease or difficulty and/or controllability of managing one’s behavior and depends on internal and external factors.

According to the TPB [26], a person develops a behavioral intention (e.g., to implement interventions that promote mental health) if he or she has a positive attitude towards the behavioral objective (e.g., improving employees’ mental health), perceives appropriate subjective or organizational norms (e.g., that employees’ mental health is important for the organization), and experiences behavioral control for carrying out the behavior successfully (e.g., a feeling of self-efficacy or sense of control when implementing mental health-related interventions). Thus, according to the TPB, one can assume that managers will actively support mental health-related organizational interventions when they believe that such interventions correspond to the organization’s norms and standards as well their personal attitudes, and when they believe that they have control to influence and change working conditions [27]. Although the TPB has not yet been frequently applied in organizational contexts, a few studies have applied the TPB in the context of health-related organizational interventions [27,52,53] or change management [35].

We use the TPB as an empirically validated framework for our qualitative systematic review and describe the methodological approach below.

## 2. Methods

### 2.1. Design

The study’s methodological approach follows the conventional steps for systematic reviews and meta-syntheses of qualitative data [54]. Accordingly, the review process was conducted in five steps: systematic literature search, selection of studies, appraisal of included primary studies, data extraction and data synthesis, and applying the framework synthesis approach. The five steps are explained in more detail below.

### 2.2. Systematic Literature Search

Studies were identified by the first author by searching the following databases: PubMed (NIH), PsycINFO and Scopus (results up to September 2021). The search was restricted to original articles published in peer-reviewed journals in English and German. We did not make any restrictions regarding year of publication. A combination of search words (e.g., MeSH) and free-text words was used. The bibliographies of the selected articles were reviewed and checked for further matches. Our search strategy was based on the PICo scheme, which is particularly recommended for qualitative reviews and fully comprehensive searches [55]; it consists of the following elements: P = Population, I = Phenomenon of Interest, Co = Context [56]. The target population of our study are managers at different levels of the organizational hierarchy. As this review was conducted as part of a larger study in a hospital context, we additionally included chief physicians, senior physicians and senior nurses. Our Phenomenon of **I**nterest is managers’ action-guiding mental models, which we captured in the search string based on the three predictors identified in the TPB. Health-related organizational interventions provide the **Co**ntext for our research question. In all databases, we searched for synonyms and generic or higher-order terms for each element of the PiCo scheme and included them in the search strings. The search strings are reported in Table A1.

### 2.3. Selection of Studies

The inclusion and exclusion criteria used for study selection were defined a priori by 2 authors (MG, AM) based on the target population, setting, topic area, methodology, language and study design (see Table A2).

We included studies with upper and middle managers as participants. The studies had to investigate managers’ perceptions of mental health-related organizational interventions. Such interventions could be part of multifaceted occupational or workplace health programs, programs to increase organizational mental health, healthy leadership programs or occupational health guidelines.

We excluded articles that address other populations such as employees, or that focus on managers’ perception of interventions promoting behavioral or physical health. We also excluded articles evaluating health-related organizational interventions from an economic perspective or investigating managers’ views of chronically ill employees.

Articles were included when all inclusion criteria were fulfilled. One author (MG) first evaluated all records based on their titles and abstracts. In ambiguous cases, the papers were discussed with one other author (AM) and the full texts were consulted. In a second step, all selected papers were independently reviewed by two authors (MG, AM) based on their full texts, and again, ambiguous cases were discussed. 

### 2.4. Appraisal of Primary Studies

We used the Critical Appraisal Skills Programme (CASP) tool for quality assessment in qualitative research [57,58]. The CASP checklist is recommended for the meta-synthesis of qualitative studies [59]. The appraisal checklist contains ten questions (see Table A3) covering three broad issues that need to be considered when appraising a qualitative study as proposed by the Critical Appraisal Skills Programme [58]: What are the results? Are the results of the study valid? Will the results help locally? We examined whether each study fulfilled the criteria for each of the ten questions by selecting ‘yes’ (√), ‘no’ (X) or ‘can’t tell’ (?) for each criterion. The scoring depended on how many questions could be answered ‘yes’ (each ‘yes’ = 1 point). A score of seven or more (maximum ten) was categorized as ‘very good quality’ [60]. Any disagreements in scoring were resolved through discussion.

### 2.5. Data Extraction and Synthesis

Full data extraction and synthesis of the articles included in the systematic review were conducted by the first author (MG). In the extraction, we considered both first-order constructs (quotes by participants) and second-order constructs (researchers’ interpretations, statements, assumptions and ideas) as data [54].

We adopted the framework synthesis approach, which examines whether other theoretical models and components can be identified that can complement our selected framework model [61,62,63,64]. Framework-based synthesis is used to answer applied (policy) questions. In this process, researchers select a priori a theoretical model that is appropriate for the research question and use it as the basis for their initial deductive coding framework. This framework is then modified inductively based on the content of the studies reviewed, so that the final product is a revised framework that may include modified components as well as new components that were not included in the original model [64]. The TPB model [26], adapted to the context of occupational health and safety measures [27], served as the basis for our framework synthesis. We applied the TPB because it allows us to capture managers’ action-guiding mental models of health-related organizational interventions and thus make predictions about managers’ behavior in implementing these interventions. However, we do not take the framework model as a static structure, but rather as a starting point and orientation for our data analysis. The framework model thus helps us to systematize the existing state of knowledge and simultaneously explore new aspects.

We considered the following framework synthesis steps identified by Pope et al. (2000): (1) “Identifying a thematic framework”: We defined the TPB model a priori as our framework model, which we specify below based on the context and objective of our research. (2) “Familiarization”: We created an overview of the following main characteristics of the included articles: author, title, year, study design, sample, aim and research question(s), theoretical framework, key findings. (3) “Charting”: We created an overview of other theoretical models and components used in the selected articles that appear relevant to our research question. Based on this overview, we decided whether we should extend or modify our TPB-based framework model with components of other models (e.g., as subcomponents or additional main components) and which components, if any, from the other models overlap with our framework components. (4) “Indexing”: We read each study in depth and deductively coded the text passages that include our extended framework components. Additional relevant results that relate to the research interest were coded inductively as new components. The software MAXQDA 2018.1 was applied for this step. (5) “Mapping and interpretation of results”: We condensed and interpreted the results according to the research questions and the components of the extended framework model.

### 2.6. Specification of the Framework Model Based on the Theory of Planned Behavior (TPB)

Our framework synthesis is based on the TPB model [26]. As previously discussed, the TPB model posits that any behavior can be predicted using the following three main components: attitude, subjective norms and PBC. For the context of occupational health and safety measures, Genrich et al. (2020) adapted the TPB components to improve understanding of which predictors influence managers’ health-promoting behavior. We applied their specification of the TPB model as our framework synthesis. Managers’ attitude regarding the importance of mental health-related organizational interventions is described via three subjective beliefs as subcomponents: “Belief in importance” refers to managers’ belief in the importance of employee mental health or health promotion. “Belief in role” refers to beliefs about whether promoting employee mental health is the responsibility of managers. “Belief in outcome” refers to the manager’s belief that health-related organizational interventions will have a positive or negative effect. Due to the organizational context of our study, we describe subjective norms as “organizational norms” (like organizational culture) that influence managers’ behavior with respect to health-related interventions. The last component of the TPB, “perceived behavioral control” refers to the manager’s experience of self-efficacy and/or sense of control regarding the implementation of health-related interventions due to internal or organizational facilitators and barriers.

## 3. Results

### 3.1. Description of Selected Studies

The results of our search strategy are shown in the PRISMA flowchart in Figure 1, including the recommended reporting elements for systematic reviews and meta-analyses [65]. A total of 8933 articles were identified at first due to the highly broad search strategy. A total of 98 abstracts (and some full texts) were checked for eligibility, and 72 were excluded. In total, 17 articles describing 17 qualitative studies and 3 quantitative TPB-based studies met the final inclusion criteria. We decided to focus our review on the 17 qualitative studies. Since we believe the 3 quantitative studies support the choice of the framework model, we include them additionally in the discussion and conclusion. 

The selected studies were conducted in various European countries, the US and Australia. The business sectors included are diverse, but 8 of the 17 studies involve managers from health or social organizations. The other sectors include manufacturing and service, finance, building and construction, aviation, higher education, information technology, trade unions, media, and the public sector. Twelve studies describe the perspective of middle or senior managers, while five studies describe the perspective of top management, including employers. Table 1 provides an overview of the main characteristics of the included qualitative studies. 

### 3.2. Quality Appraisal

The quality of the included articles varied from six to ten points. Fourteen studies had ‘very good’ quality, with scores equal to or higher than seven. Three further studies scored five or six. A qualitative design was appropriate for all studies, and each study clearly stated the aims of the research. The most common reason for deducting points was that the relationship between researcher and participant was not sufficiently clarified. Moreover, for some articles it was not possible to evaluate whether the recruitment strategy was appropriate for the research objectives. Due to the exclusively good or very good quality ratings, all articles were considered in the further analysis. Table A3 provides an overview of the results of the quality appraisal.

### 3.3. Data Extraction and Synthesis

Table 1 presents an overview of the main study characteristics of the included articles with respect to the following: author, title, year, study design, sample, aim and research question, theoretical framework and key findings.

### 3.4. Overview of Other Theoretical Models and Approaches

Six of the seventeen studies follow a deductive approach and explicitly refer to different theoretical models (see points 1–4). Eleven studies use an inductive approach to develop theoretical approaches, core themes, or models. We summarize these in point 5. One study (Genrich et al., 2020) adapted the TPB model to health-related organizational contexts, which we described in the methods section as the TPB-based framework model that formed the basis for our research and take up here as a basis for comparison with the other theoretical models.
Horstmann & Remdisch (2019) examine drivers and barriers related to the four components of the health-specific leadership model (“health value”, “health awareness”, “role modeling”, “health behavior”) [66] at the leader, employee and organizational level. “Health value” refers to managers’ interest in their own health and the health of their employees. “Health awareness” refers to managers’ awareness of their own and their employee’s health status, their employees’ work demands and resources, and possible interventions. “Role modeling” refers to managers’ self-care, which can serve as an example to employees and encourage healthier behavior [67]. “Health behavior” refers to all of managers’ health-related actions, including changing work conditions or job design (i.e., health-related organizational interventions), which is what should be explained by the TPB components in our framework model. Similarly, Efimov et al. (2020) address the importance of “health awareness” and “health value” using the health-oriented leadership model [68].Based on the social-ecological model (SEM) for health promotion [69], Zhang et al. (2016) argue that barriers and supporting factors for implementing health-related organizational interventions can be described on four different levels: the intrapersonal, interpersonal, organizational and corporate level. The study distinguishes between the corporate level and the organizational level, as the study is based on a company with several subsidiaries. “Intrapersonal” refers to characteristics of the individual, such as knowledge, attitudes, self-concept and skills. “Interpersonal” refers to formal and informal social networks and social support systems. “Organizational” refers to formal and informal rules in the organization, and “corporate” refers to the relationships among different subsidiaries. The SEM assumes that the different levels interact with each other, mutually affecting the effectiveness of organizational interventions [47,69].According to the health action process approach (HAPA) [70], applied by Schulte & Bamberg (2002), managers are likely to engage in health-related behavior when they experience their own health as threatened, when they know about appropriate and effective interventions, when they believe that these interventions can be carried out successfully (self-efficacy), when the intention to do something to improve is translated into concrete action planning, and when hindering factors (barriers) are low and promoting factors (facilitators) are high.The diffusion of innovations model [71], applied by Kalef et al. (2016) describes five characteristics of an innovation (including a health-related organizational intervention) that can affect implementation: (1) “Relative advantage”: the perception of whether the measure improves the current situation. (2) “Compatibility”: the degree to which the measure is perceived as consistent with the values and needs of potential users. (3) “Observability”: how visible the results of the measure are to others. (4) “Trialability”: the extent to which the innovation can be tested. (5) “Complexity”: whether the measure is perceived as easy or difficult to understand and apply.Five of the eleven inductive studies identify critical individual or organizational conditions (as barriers or facilitators) for the implementation of health-related organizational interventions [72,73,74,75,76]. Quirk et al. (2018) identify three levels of barriers and facilitators: interpersonal, cultural and policy. Another five studies focus on the role or responsibility of managers in the context of promoting employee’s mental health [77,78,79,80,81]. One study focuses on responsibility for mental health promotion from an ethical point of view [81].

Table 2 provides an overview of how these additional theoretical approaches were related to the TPB-based framework model. Overall, the table shows that the dimensions of the identified theoretical approaches (points 1–5) can be mapped onto the TPB-based components. We will briefly illustrate this with the diffusion of innovation model (4): when managers perceive a “relative advantage” of implementing an organizational measure to improve mental health, this corresponds to the “belief in outcome” attitude from the TPB. Similarly, “compatibility” in the diffusion of innovation model corresponds to organizational norms in our framework model. Finally, the “observability” of the intervention results to others, the ability to try out the intervention, and the low “complexity” of the intervention can be seen as facilitators that affect managers’ perception of behavioral control.

As almost all theoretical aspects of the other models were captured in the specified TPB-based framework model, we stuck to this structure in the in-depth content analysis. Only for the component of perceived behavioral control did we focus on perceived upstream facilitators and barriers in the in-depth content analysis, as we believe these better represent the content of the studies.

### 3.5. Mapping the Results

#### 3.5.1. “Belief in Importance” of Mental Health-Related Organizational Interventions

Eleven studies address this sub-component. Ten of these find that the interviewed managers are aware of the importance of employees’ mental health and mostly attach great importance to improving the status quo. Some managers explained their concern for employees’ health not only as a personal attitude, but also as a duty of an employer [68]. In the study by Kunyk et al. (2016), managers saw mental health and safety as the core of all health issues at work, with serious costs in the case of illness, and as an issue for the next generation. Some studies report that managers are also aware that psychosocial job characteristics influence employee mental health: Havermanns et al. (2018) find that managers identify a need for attention to different determinants of work stress, e.g., job demands, autonomy/control, clarity about work tasks [74]. Similarly, Genrich et al. (2020) found that the topic of mental health was highly relevant to hospital managers, but they were unable to take it into account in their work routines [27]. Some managers consider employee health important, but see other issues at work as more important, especially the organization’s profitability [17]. Only one study reported a lack of awareness among junior healthcare managers about employees’ work stress and how to manage stress effectively and proactively [75]. The study further indicates that this attitude was related to organizational norms, which they found to be characterized by an acceptance and expectation of work stress.

#### 3.5.2. “Belief in Outcome” of Mental Health-Related Organizational Interventions

Eleven of the studies provide information on managers’ belief in the benefits of employees’ mental health or health-related organizational interventions at the workplace. Managers take two perspectives on this issue: the economic-related perspective, which focuses on increased profitability, and the employee-related perspective, which emphasizes employee health as a value in itself. We found that the economic or more functional perspective predominates [27,67,73,75,78,82]. Managers associate healthy employees and a healthy working environment with aspects such as improved employee performance, a good corporate image, loyalty, recruitment and retention, accessibility and safety as well as reducing costs, conflicts and other issues [72,73]. Additional reported expectations regarding positive consequences of good mental health are: employee motivation and risk reduction [75], reduced sick leave [17,72], higher productivity, increased collegiality and the ability to maintain a work–life balance [72,79]. Pescud et al. (2015) report that a substantial number of managers perceive an association between healthy employees and the company’s business goals, while only few managers in that study reported little or no association.

Six studies also provide insights into managers’ attitudes towards health-related organizational interventions or job characteristics that they believe have an impact on employees’ mental health, as a value in itself [27,73,74,75,82]. Hospital managers particularly mention interactional or socially supportive approaches to health promotion, e.g., assisting with the completion of tasks or showing appreciation; approaches such as redesigning work tasks or work processes are mentioned less often [27]. We found various indications that managers believe that health-related organizational norms support the implementation of health-related organizational interventions or healthy job characteristics [27,47,73,74,75,82]. We report on these assumed associations in more detail in the context of organizational norms.

Three studies highlight managers’ skepticism by reporting their critical attitude towards a corporate policy to promote health: Junior managers in hospitals associated stress exclusively with individual behaviors and did not see the value of health-related organizational interventions [80]. In addition, in two studies, some executives did not believe that employee health and productivity are related [79,80]. These managers see a company’s profitability as the basis for ensuring employees’ well-being—not vice versa. Profitability is seen as the essential characteristic of a healthy company [17]. Although healthy employees are often seen as an important resource for the company, the need to introduce health-related organizational interventions was not often mentioned by participants [79].

#### 3.5.3. “Belief in Role” in Implementing Health-Related Organizational Interventions

Fourteen of the seventeen studies present findings on managers’ understanding of their own role in and sense of responsibility for promoting employee mental health. Across studies, managers describe very different understandings of their roles and responsibilities: On the one hand, there are managers who believe that employees themselves are responsible for their own health or stress management in the workplace [17,80], while other managers feel responsible for implementing or supporting health-related organizational interventions [8,75,78].

Respondents in Schulte & Bamberg’s (2002) study were unanimous in their opinion that the employee him- or herself is primarily responsible for his or her health, with the company responsible for ensuring that employees’ health is not negatively affected during their work hours. Surprisingly, the same respondents prefer mental health-related organizational interventions to behavioral interventions, as it is not the responsibility of the company to intervene in employees’ personal sphere [17]. Some managers also distinguish between responsibility for employee health and responsibility to create healthy working conditions [68]. They do not feel responsible for the health of their team members but do assume responsibility for creating healthy conditions. Junior medical mangers in the study by Rodham & Bell (2002) did not feel that the organization should take any responsibility for work stress management [80].

On the contrary, other managers assume a much higher level of responsibility. They believe that they have a key role in the development and implementation of workplace health promotion activities (e.g., to identify risks and maintain or improve their employees’ health), both in terms of the working environment they create and their overall management of employees [8,72,75,78]. Some managers see this as their key function and at the same time perceive a conflict with economic pressures [17,27]. Our analysis also shows that there is a degree of uncertainty among managers in some organizations about their role: For example, managers feel responsibility for communication about work stress, but are unsure whether they or the employee should begin the dialogue [74]. Some managers state that they feel responsible for helping to solve employee problems, even when doing so felt burdensome [8,27,75].

Summary regarding managers’ behavioral beliefs:

Managers are aware of the importance of promoting employees’ mental health and mostly attach great importance to improving the status quo in terms of work performance. However, managers feel responsible for the topic to very different degrees: some do not feel responsible at all, some feel insecure, and others believe that they have a key function in the implementation of mental health-related organizational interventions.

#### 3.5.4. Perceived Organizational Norms

Ten of the studies addressed managers’ perceptions of organizational norms. Many managers perceived that mental health plays only a subordinate role in their organization’s norms or that existing norms actually hinder health promotion. Three studies reported that managers perceive less engagement or presence and a lack of communication about employee mental health from upper management [8,27,74], a culture of fear [74], a top-down mentality in healthcare organizations with the feeling that, ironically for an organization dedicated to health, “it’s not okay to focus on employee health”. Here, the hypothesis was put forward that the examined organizations have little or no receptivity to implementing mental health and safety strategies [73]. Three other studies, also carried out in the healthcare sector, support this perspective: Managers perceive that organizations within the healthcare sector are traditionally viewed as a service that cares for and invests in services for patients, not in its staff [27,76]. Junior hospital managers perceive a culture of acceptance of work stress, combined with a lack of awareness on how to effectively and proactively manage it [80]. As reported above, these young managers do not believe that the organization should take responsibility for managing work stress, nor do they have a strong awareness of it [80].

Although the majority of studies do not explicitly focus on the relationship between organizational norms and managerial behavior, there are some indications that health-related organizational norms affect managers’ behavioral intentions to implement health-related organizational interventions. Schulte & Bamberg (2002) noted that the majority of managers who expressed the intention to actively support health-related organizational interventions came from a company where the implementation of a comprehensive health policy (as an organizational norm) had previously been assessed as positive. Larsson et al. (2016) report that guidelines for creating an orientation towards workplace health promotion support the implementation of health-related organizational interventions in different departments of municipal organizations (e.g., childcare and education, transportation, and urban planning).

In the studies by Kalef et al. (2016) and Kunyk et al. (2016), managers desire and anticipate positive effects of their organization introducing health-related standards to promote health-related organizational norms. These managers believe that such standards may provide a guiding framework for greater awareness and understanding of mental health in the workplace. Such standards are seen as a guideline or tool for promoting employees’ mental health and clarifying employer responsibility [82]. From managers’ perspective, health-related norms should focus on prevention and be strategically oriented, i.e., incorporated into existing corporate goals and visions [73,75,76,82].

Summary for Organizational Norms:

Our analysis shows that managers’ perceptions of organizational norms vary. Many managers perceive organizational norms as having little to no connection to health or employees, but rather focused on improving organizational performance. Organizational norms seem to have an impact on managers’ sense of responsibility and health-oriented leadership behavior.

#### 3.5.5. “Perceived Behavioral Control”: Barriers to the Implementation of Health-Related Interventions

Organizational level: A lack of organizational resources is most frequently cited by managers as a barrier to the implementation of mental health-related organizational interventions. These include lack of time and personnel and financial constraints [27,67,72,74,75]. A total of 10 of the 15 qualitative studies mention at least one of these barriers. Lack of time and personnel hinder managers (as well as employees) from implementing (or participating) in health-related organizational interventions [47,67]. Shift work and remote workplaces seem to increase these difficulties, because they make a common dialogue within the team more difficult [27,73,76]. Healthcare managers in particular mentioned high fluctuation in teams and lack of continuity as additional factors hindering the implementation of mental health-related organizational interventions [27,67,76].

Managers perceive time and role conflicts as further barriers, particularly when they have high workloads and have to meet a variety of challenging job demands, tasks and functions [8,46,72], which often relate to economic goals [17,74]. For example, in the study by Larsson et al. [46], one manager described that the development of action plans and implementation of health-related organizational interventions failed because managers had to perform other tasks.

Interpersonal level: Two further barriers to the implementation of interventions to promote health are communication problems [47,74,75] and lack of engagement of important stakeholders [17,27,47,67], especially upper management, but also employees themselves [74]. For example, supervisors stated that employees must be “ready for change” to avoid work stress. This individual readiness for change was described as the awareness and recognition that work stress is a problem and willingness to participate in (and not resist) the prevention process [74]. Communication problems are mentioned when working groups planning health-related organizational interventions mainly keep their dialogue within the group instead of involving employees further down in the organizational hierarchy [47]. In addition, it becomes difficult for managers if employees do not communicate clearly and honestly about their health issues and work stress [74,75]. On the other hand, managers state that it becomes difficult when such openness leads to stigmatization of employees [74]. Other priorities among upper management, competitive corporate objectives [17,67] and a top-down decision-making (hierarchical) structure [27,47] are further difficulties that executives reported in several studies.

Individual level: As individual barriers, managers report insufficient knowledge about mental health issues and supportive leadership [27,73,74]. Managers of small organizations describe difficulties in combining different roles [75], while other managers describe a feeling of limited self-efficacy in terms of a pervasive and overwhelming feeling of powerlessness [27,73].

#### 3.5.6. “Perceived Behavioral Control”: Facilitators for the Implementation of Health-Related Interventions

Organizational level: From managers’ perspective, health-related organizational interventions should be embedded in a coherent, strategic approach or a workplace culture [72] that is supported at all levels of the organization, from upper management to bottom-line employees [72,76]. Connectivity to existing structures is seen as helpful [73,82], the same applies to integrating health promotion into organizational development [72]. Top-down initiation of interventions and support from top management are essential from the managerial perspective [27,76,82]. It is perceived as helpful to have clearly defined management responsibilities and resources (i.e., time, finances) for workplace health promotion, and a steering group to catalyze and coordinate health-related organizational interventions [47,72,76]. Managers of an organization should pull together and serve as role models in their departments, which can in turn influence the organizational culture [72]. Health-related organizational interventions are described as easier to implement if they can be introduced in small, simple steps. Health-related organizational interventions should be relevant, understandable and flexible enough to be adapted to specific working conditions (company size, location, sector) [73]. 

Interpersonal level: Functioning and effective communication of health-related organizational interventions at both the organizational and team level is seen as another important facilitator [73]. The added value and benefits of interventions must be clearly articulated by top management and be oriented to the organization’s needs and current challenges [73]. Regular informal [75] or open and ongoing dialogue within the team [8,27,67,68,72] or an annual employee survey [46] are considered helpful for promoting effective communication. Close direct contact with employees is seen by some managers as helpful in anticipating problems [74,75]. A positive team climate is perceived as facilitating the implementation of health-related organizational interventions, as it promotes mutual support, the assumption of responsibility, and a trusting dialogue within the team [75]. 

Individual level: Some managers refer to health-specific knowledge and competencies, i.e., how to design healthy workplaces, the “right” attitude [72] and a proactive “hands-on” mentality, i.e., determination, willingness to take risks, flexibility, persistence and pragmatism [67]. Others perceive that they can control the workload by setting priorities to reduce their employees’ stress [27].

Summary for Facilitators and Barriers (Control beliefs): 

Mainly organizational-level barriers hinder managers in implementing mental health-related organizational interventions. Lack of time, personnel and scheduling problems are frequently mentioned, followed by a lack of financial resources and communication problems. Insufficient knowledge about mental health issues and uncertainties about how to promote employees’ health reinforce these barriers. A strategic approach for health that is initiated and promoted by top management and provides the other management levels with a scope for action, as well as a good team climate, are seen as central facilitators. 

#### 3.5.7. Identified Background Factors for Managers’ Mental Models

In our inductive analysis, we were able to identify additional components that we believe may act as background factors for leaders’ mental models and thus be relevant to the implementation of mental health-related organizational interventions.

Pescud et al. (2015) report that managers from metropolitan areas, especially white-collar workers, tend to attach greater importance to their employees’ mental health, and women do so more than men. Managers from rural areas tended to focus more on safety issues related to preventing accidents and injuries, with mental health “only” in second place [79]. If representative of the wider organizational community, health-related organizational interventions are more likely to be promoted in larger businesses that have clearly distinguished such measures from their legal responsibility for health and safety [79].

Schulte & Bamberg [17] highlight region-specific differences in the perception of managers from Germany and Scandinavia. Understanding of corporate responsibility for employee health seems to be particularly influenced by the national cultural framework. Schulte & Bamberg [17] describe German managers’ skepticism about a standardized health policy that does not fit into existing organizational norms and is predestined to fail. They note that some of these managers see health-promoting approaches as incompatible with organizational norms regarding profitability and increased performance. In contrast, due to positive previous experiences with feasible, practical company and leadership guidelines in other areas, most Scandinavian managers seem to be more optimistic.

Pescud et al. (2015) conclude that employers from smaller workplaces were more likely to feel personally responsible for their employees’ health, particularly their mental health, than employers from bigger companies. This is attributed to the closer personal relationship between supervisors and employees in smaller companies. Landstad et al. (2017) conclude that small organizations are particularly suited to focus on designing mental health-related organizational norms that serve as the basis for a healthy workplace. However, whether they can make use of this potential to further develop mental health-related organizational interventions is more questionable than in more bigger organizations due to limitations in framework conditions (finances, personnel, know-how) and high demands [81,83].

Two studies indicate that managers’ hierarchical position has an impact on their attitudes or behavioral beliefs towards health-related organizational interventions [8,27]. While senior managers see themselves as deciding whether a given intervention should be implemented or not, middle managers feel responsible for ensuring that the intervention is successfully implemented in practice [8]. 

## 4. Discussion Regarding of Managers’ Action-Guiding Mental Models

This article contributes to a better understanding of managers’ action-guiding mental models regarding mental health-related organizational interventions by systematizing and theoretically classifying the hitherto fragmented literature in this field from the perspective of the theory of planned behavior (TPB). This allows theoretical sound conclusions to be drawn about whether managers may be potentially willing to promote or support organizational interventions.

We believe this perspective can also enrich existing research on general organizational change processes. In organizational change management research, it is undisputed that leadership is one of the most important factors for the successful implementation of change processes [14,30]. There is an extensive literature examining leaders’ roles, responsibilities, and characteristics in relation to change [84,85,86]. However, this research is primarily descriptive and based on observations by managers, subordinates, or peers regarding managers’ knowledge, skills, abilities, and effectiveness [14,84]. Our study results complement existing research by adding managers’ internal perspective, which should be paid more attention with respect to engaging managers in organizational change processes [14].

In our paper, we described psychological action-guiding factors that can inform future research on organizational interventions and help to explain why managers do or do not support organizational change. In the run-up to a change process, these factors can provide important information for change agents on the extent to which specific measures (e.g., additional information, participation in goal setting, training) can convince managers to support change.

First, distinguishing the attitude component proposed by Genrich et al. (2020) into three major underlying beliefs (“belief in importance”, “belief in outcome”, “belief in role”) contributes to a more differentiated understanding of how managers’ attitudes are related to their actions towards organizational interventions. For example, our findings suggest that positive attitudes toward the outcome of an organizational intervention alone (in this case, employees’ mental health) may not be sufficient to support that intervention; in addition, managers must also see themselves as responsible for contributing to this outcome, (i.e., belief in role). As previous research has shown, differentiating attitudes towards particular behavioral contexts is important because global attitudes are generally poor predictors of specific behaviors (e.g., the implementation of health-related organizational interventions) [87]. Beliefs associated with the specific behavior of interest have a more direct influence that may exceed the impact of global attitudes [87]. We made such a distinction with respect to implementing mental health-related organizational interventions, which should be taken into account when recruiting or motivating managers for health-related organizational interventions.

Our results show that managers’ “belief in role” and sense of responsibility for promoting employees’ mental health vary. Our study provides explanations for these differences. The studies in which managers had a more reserved attitude were published comparatively earlier (in 2002), and we assume that managers may have recently become more open to the topic. While this assumption needs to be confirmed in follow-up studies, various reports indicate increased acceptance among organizations that mental health is an important issue. For example, more recent studies report an increase in the prevalence of workplace health programs, albeit still at a low level [88,89]. Another explanation comes from Downey and Sharp [90], who used the TPB perspective. They showed that managers who are under greater financial pressure report less moral responsibility for promoting employees’ health than others. This group of managers appears to be more interested in the functional outcomes (i.e., increased productivity, less sick leave) of health promotion. Attitudes thus seem to be influenced by organizational conditions, which we discuss in depth below when considering organizational norms.

In addition, our results also reveal that managers may have different beliefs about the causes and instrumental purpose of the outcome. These different beliefs may in turn be associated with different behaviors. For example, if managers believe that a company’s profitability is a prerequisite for employee health (see Schulte & Bamberg, 2002) they may focus their attention primarily on the former rather than on health promotion in the narrower sense. Another assumption is that managers may not be aware of work-related risks to employees’ mental health [91] and health-promoting work design approaches (e.g., fostering autonomy, participation) and need to catch up here. It is possible that different persuasive approaches may work for different managers, depending on whether they view employee health as a means to an end or as a value in itself.

Although most included studies suggest that managers have positive attitudes toward employee mental health, they do not always see themselves as responsible for promoting it. Managers also seem have very different theories about what contributes to employee mental health. Consequently, it is precisely here that companies need a clear clarification of roles and an educational approach regarding the causes of mental health. It can be expected that managers will be more likely to support health-related interventions if they consider mental health to be an important issue, if they believe that mental health-related organizational interventions achieve positive outcomes (also economically) and if they believe that they are responsible for employees’ mental health. If only one or two components are weak, it is helpful to concentrate persuasion efforts on those components.

Second, we found evidence that organizational norms may play a crucial role, as they appear to be interrelated with other components of managers’ action-guiding mental models. Our findings indicate that organizational norms (e.g., in the form of health-related organizational standards or leadership guidelines) may have an effect on managers’ sense of responsibility and affect their attitudes toward organizational interventions. Accordingly, a lack of or contradictory organizational norms (e.g., acceptance of work stress, focus on performance) can lead managers to not feel responsible or to role conflicts. This relation is consistent with the perspective of organizational role theory, which assumes that roles in organizations are associated with specific social positions (e.g., the leadership role) and are generated by normative expectations (e.g., reaching economically oriented goals). Norms can vary and reflect both organizations’ official requirements and informal group pressures, not always in line with each other, which can in turn create role conflicts [92].

Organizational norms may also be related to perceived behavioral control in the implementation of health-related organizational measures. For example, positive organizational norms (e.g., health-related standards, top management support for health-related goals, open communication about health-related issues) are perceived as facilitating, as they give managers guidance regarding their responsibilities and role in implementing health-related policies and measures. This normative component can be connected to the concept of psychosocial safety climate (PSC) [93]. The PSC refers to the organizational climate for employees’ mental health, well-being and safety and encompasses organizational policies, practices and procedures designed to protect employees’ psychological health and safety [94]. PSC is described as a “cause of the causes of work stress” and thus a primary risk factor in organizations [95]. Several studies provide evidence that perceived organizational norms or a psychosocial safety climate impact health-promoting leadership behavior [52,96]. Wilde et al. (2009) highlight the importance of organizational norms (described as a “culture of healthy leadership”) as a direct predictor of health-related leadership behaviors. Another study provides evidence that a positive PSC is an indicator of supportive leadership behavior that is visible on a daily basis [96]. Our findings suggest that in the case of marginal organizational topics such as mental health promotion, organizations should start from the top down with the goal of establishing and disseminating norms and standards in order to engage managers’ behavior. 

Third, we were able to reveal that managers’ perceived behavioral control is related to facilitators and barriers on multiple levels (organizational, interpersonal, individual, interpersonal). As already mentioned in relation to organizational norms, organizational-level factors have an instrumental function (for example, by providing resources), and also help orient managers as to which actions are desirable or undesirable and can thus help to avoid role conflicts. At the interpersonal level, open, trusting, and continuous communication helps managers identify relevant stressors at work and foster employees’ participation in the development of steps for improvement. Thus, we confirm the existing literature on the importance of communication in change processes [30,84]. At the personnel level, knowledge of the relationship between working conditions and employees’ mental health is central for managers to feel self-efficacious in making working conditions healthier.

It can be assumed that barriers at the different levels influence each other, since individuals are embedded in team structures, and teams are in turn influenced by organizational structures [97]. Although identifying barriers at the various levels already seems helpful as a way to identify more precise approaches to increasing managerial behavioral control, it would be helpful if future research could provide more insight into the interrelationships between the different levels. For example, it remains an open question whether the removal of barriers at the organizational level is automatically accompanied by an improvement at the team or individual level, or whether it can also lead to new barriers at these levels.

With respect to mental health-related organizational interventions, managers often seem to perceive more organizational barriers than facilitators. This might be one reason why they seem to concentrate on individual and team-related interventions, as they perceive a greater sense of self-efficacy and controllability to promote employee health at these levels [27]. The successful implementation of mental health-related organizational interventions, on the other hand, seems to be more dependent on the availability of organizational resources (time, personnel, money) and the above-mentioned normative facilitators. Another possible conclusion could be that many managers do not know how to design working conditions in a healthier way. Hence, it can be assumed that many aspects of mental health-related organizational interventions suggested by established theories, such as increasing job autonomy, optimizing job demands, and eliminating obstacles in work processes [98], are not or cannot be considered by managers as a way of maintaining their employees’ health. There seems to be a gap between research and practice, which is discussed in work and organizational psychology at present [99]. Low-threshold approaches need to be developed in order to make knowledge available to practitioners in organizations in a way that allows them to integrate this knowledge into work design concepts.

Fourth, we identified potential background factors on different levels that might influence managers’ perceptions and behavior: Particularly, managers’ gender and age, hierarchical position, the size and sector of the company, and the country and cultural framework might play a role. In the literature, we found evidence that older people and females have more positive attitudes towards psychological help-seeking and are more likely to seek psychological help than men and younger people [100]. Women’s openness to acknowledging mental health problems is consistent with research suggesting that they are more likely than men to recognize and label emotional distress [100,101]. Another study examined the influence of age and gender on attitudes toward mental health and found that middle-aged and older people (above age 40) had less stigmatized attitudes toward mental illness than 16- to 18-year-olds [102]. Individuals in the 40+ age group are probably more likely to know people with a mental health problem than younger people, which might reduce stigmatizing attitudes and increase awareness of the importance of mental health promotion [103], while younger people tend to evaluate mental health problems as a personal failure or weakness rather than a valid health problem [104]. Sector-specific differences might be traced back to the gender distribution in the organization. As a rule, manufacturing sectors are dominated by men, while business services are more dominated by women. Smaller companies generally have fewer staff and financial resources, so that much lower-threshold health promotion programs should be implemented.

These findings suggest that these background factors are further predictors of managers’ mental models and should be considered with respect to the implementation of psychosocial organizational interventions. TPB alone does not address where managers’ mental models (as behavioral, normative, and control beliefs) that influence their behavior come from [105]. We therefore suggest that the TPB model should be framed within a broader context and that the introduction of mental health-related organizational interventions requires a differentiated strategy that is sensitive to company demographics, business sector, size, and location or cultural background.

All in all, our study shows that the theory of planned behavior is suitable as a framework model for identifying aspects that guide managers’ actions and that should be considered in order to understand and foster health-promoting behavior by managers. A more in-depth exploration of this topic could catalyze further research on the implementation of mental health-related organizational interventions as a contextual change process. 

## 5. Limitations and Directions for Future Research

First, we would like to point out that we identified only a relatively small number of 17 qualitative studies on which the discussion of our findings is based. Furthermore, the 17 identified studies originated from different countries (USA, Australia, European countries), each with different backgrounds in workplace health promotion. The results and the resulting discussion and conclusion should therefore be taken with caution. In our opinion, the rather small number of studies may be a sign that the research focus discussed here has not been adequately addressed in previous research. Future qualitative research should therefore further consider the perspective of leaders’ mental models in relation to healthy leadership behaviors. Our systematic review can provide as a basis for the interpretation of this future research.

Second, selection bias effects cannot be ruled out. Often, managers are interviewed who are already interested in the topic. In our view, this selection effect might particularly bias the attitude component. Moreover, managers might exhibit socially desirable response behavior when asked about their attitudes toward mental health. For example, Hasson et al. (2014) emphasize in their study that managers can in practice rarely fulfill all the roles and tasks they report. This highlights the usefulness of also obtaining an external perspective on managers behavior in future studies, e.g., by interviewing subordinates or other stakeholders in addition to managers. For example, future studies could examine whether the perspectives of subordinates or other stakeholders correspond with those of managers. In our study, we were less interested in obtaining a representative picture of the prevalence of these components than in revealing potentially relevant aspects that play a role in managers’ thinking in the first place.

Third, we reported evidence that organizational norms affect managers’ behavioral beliefs and perceived barriers and facilitators, which in turn influence their perceived behavioral control. However, our qualitative approach is not able to statistically validate these relationships. Future quantitative research could investigate whether the described interactions can be empirically validated.

Because we did not specifically address background factors, we do not assume that we were able to identify all possible background factors in this study. For example, it is also plausible that personality traits affect leaders’ mental models. Studies examining the personality of transformational leaders [106] provide initial evidence for this.

We recognize that all qualitative research is contextual; it takes place between two or more people at a given time and place [107]. In the quality assessment, we noticed that not all studies transparently reported what they did to minimize situational or personal bias in the interview process. For example, it cannot be excluded that the in-depth questions posed differed based on the interviewers’ personal or professional experience and interests. Again, more quantitative studies are needed to validate our findings.

We propose the following directions for future research: Future studies should examine how managers’ attitudes and perceived organizational norms toward intervention objectives, as well as their perceived behavioral control, influence their actual behavior when implementing organizational interventions. Based on the results of our study, the following research questions seem to be of particular importance: How do belief in role, belief in outcome, and belief in importance interact to predict managers’ behavior? Do organizational norms predict managers’ attitudes and behavioral control? In this context, our study findings can contribute to the development of testable hypotheses, which in turn might help to not only refine the TPB but also to predict managerial behavior when implementing organizational interventions, especially mental health-related organizational interventions.

Standardized questionnaires to assess the action-guiding mental models we identified among managers should be developed. In addition to hypothesis testing, such questionnaires could be used to investigate managers’ readiness before the start of a mental health-related organizational intervention, which could in turn be used to inform preparatory training and sensitization measures.

## 6. Implications for Practice

Our extended TPB model can identify concrete starting points for increasing managers’ support for the implementation of mental health-related organizational interventions. Our findings suggest that raising awareness of the importance of mental health at work does not seem to be the top priority if one wants to encourage managers’ health-promoting behavior. Rather, it seems necessary to clarify managers’ roles and responsibilities on this issue and clarify the benefits for work-related outcomes. In other words, if you want to do something for employees’ health, you should argue in terms of business logics.

Thus, when it comes to marginal organizational topics such as mental health promotion, organizations should start from the top down with the goal of establishing and disseminating norms and standards that facilitate leaders’ engagement in implementing psychosocial organizational interventions. These norms should be reflected and specified in policies, practices, and procedures to protect the mental health and safety of all employees. For example, top management could ensure that health-related goals and performance indicators are included in existing management systems, demonstrating that economic success and employee health are not in conflict with each other. 

One starting point for top management might be to promote health-related organizational norms, for example by establishing a credible and transparent communication process regarding the importance of mental health promotion or by developing participatory strategic and operational goals and interventions to promote employees’ mental health and integrate them into existing structures [27]. Organizations should make it clear that promoting employee health and achieving economic goals are not in conflict with each other.

Moreover, organizational structures should be developed to remove barriers to and facilitate managers’ behavioral control. For example, a simple measure could be to establish participatory communication structures so that managers and employees can provide ideas for reducing stress [94]. Managers could be invited to participate in the introduction or development of health-related leadership guidelines to provide all managers with more orientation and clarity in their leadership role. Open communication on health-related issues and problems should be initiated so that the topic of mental health is no longer a taboo subject in organizations.

Beyond that, there seems to be a need to improve managers’ systematic training and involvement in mental health-related organizational interventions. Workshops could provide managers with the necessary knowledge and practical tools to promote and implement mental health-related organizational interventions [108]. Health-related expertise and practical tools from internal specialists (e.g., occupational physicians) or external health experts should be used to bring this necessary knowledge into departments and teams. We share and reinforce Kunyk et al.’s (2016) call to take into account the specific characteristics of organizations and managers’ positions before implementing health promotion interventions. For example, managers in different positions could be convinced by different arguments.

## 7. Conclusions

Our study indicates that the interplay among three aspects of mental models guide managers’ actions when implementing organizational interventions: managers’ attitudes and perceived organizational norms towards intervention objectives, as well as their perceived control in carrying out actual behavior to implement organizational interventions.

From a research perspective, this approach based on the theory of planned behavior (TPB) can help to develop further hypotheses on manager behavior that are relevant for the implementation of organizational interventions, particularly interventions to design healthy working conditions.

From a practical perspective, our results provide evidence that supportive organizational norms (e.g., health-oriented management standards; health-related organizational visions, goals and communication) may particularly help to create a common sense of responsibility among managers and foster their perceived controllability with respect to changing working conditions. Additionally, managers need more know-how as well as job resources such as time, money and personnel resources to implement mental health-related organizational interventions. Economic success and increased productivity seem to be one of the main motives for managers (and companies) to implement mental health-related organizational interventions. To convince managers of the usefulness of interventions, the relationship between good work design and productivity should therefore be highlighted.

## Figures and Tables

**Figure 1 ijerph-19-12610-f001:**
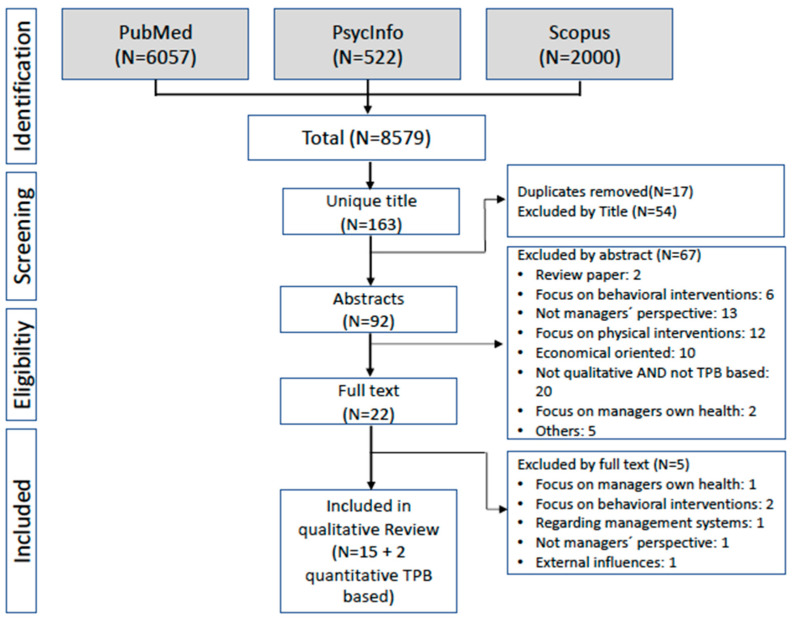
PRISMA flowchart demonstrating search strategy.

**Table 1 ijerph-19-12610-t001:** Overview of included studies.

	Author, Title, Year	Design	Sample	Aim/Research Question	Theoretical Framework	Key Findings
1	Efimov, I., Harth, V., Mache, S. (2020) Health-oriented self- and employee leadership in virtual teams: A qualitative study with virtual leaders. Int. Journal of Environmental Research and Public Health	Qualitative study: semi-structured, guideline-based telephone interviews, problem centered interviews.	13 managers (IT-sector, manufacturing industry, aerospace industry) from medium-sized and large companies.	Insights on the health awareness of leaders and on affecting enablers and hindrances to the implementation of health-oriented leadership.	Health-oriented leadership (HoL): health value, health awareness, behavior	Managers considered the value of their team members’ health to be as high as their own (employer’s duty of care, personal attitude). Different understanding of leadership roles: From a high degree of responsibility to one’s team members to a sense of being responsible not for the health of team members but for establishing healthy working conditions. Most of the managers consider an atmosphere of trust to be a basic condition for the implementation of health-oriented leadership in a virtual team. Five behaviors of health-oriented leadership were mentioned: confidence-building measures, health-oriented communication, boundary management support, conducting face-to-face meetings, and delegation of decision-making responsibilities and authority.
2	Eriksson, A., Axelsson, B., Axelsson, S.B. (2011) Health-promoting leadership—Different views of the concept. Work	Qualitative study: semi-structured interviews analyzed following the principles of phenomenography.	10 middle-managers, 4 personnel managers, 1 area manager, 1 administrative director, 2 project leaders of eight Swedish municipalities	Analysis of the different perceptions of health-promoting leadership among stakeholders (including managers) engaged in workplace health promotion. How is health-promoting leadership characterized? What are the motives for it? What critical circumstances are noticed for such a leadership?	Inductive approach. Focused on critical individual and organizational conditions for the development of health promoting leadership.	Managers describe three components of health-promoting leadership: health-promoting actions, a facilitating leadership approach, and creating a health-promoting workplace. Motives reported included instrumental results (e.g., reducing absenteeism, improving ease of hiring staff) and health benefits. Organizational circumstances (e.g., worksite environment, finances, culture of organization), qualities of each leader (e.g., knowledge, attitudes), and management facilitation (e.g., supervision, administrative support) were considered to be major determinants for health-promoting leadership.
3	Genrich, M. et al. (2020) Hospital Medical and Nursing Managers’ Perspectives on Health-Related Work Design Interventions. A Qualitative Study. Frontiers Psychology	Qualitative study: semi-structured interviews. Analyzed by content analysis.	37 managers (chief physicians, senior physicians, and senior nurses) from a German hospital.	Hospital managers’ perspectives on health-related organizational interventions.	Theory of planned behavior (TPB), regarding the predictors attitude, perceived organizational norms, and perceived behavioral control.	Most of the managers consider health-promoting organizational measures to be important. Managers disagree on the importance of organizational norms. Opportunities for implementing organizational measures are reported predominantly at the individual and team level, less so at the organizational level.
4	Hasson, H. et al. (2014) Managing implementation: roles of line managers, senior managers, and human resource professionals in an occupational health intervention Journal of Occupational and Environmental Medicine	Qualitative study: semi-structured interviews as part of an intervention study	29 interviews with line managers (*n* = 13), senior managers (*n* = 7), and HR professionals (*n* = 9) from 9 organizations in Stockholm, Sweden. Branches: higher education, information technology, trade union, media, and government authorities.	How do line managers, senior managers and HR specialists perceive their own and each other’s roles and tasks and the possibilities for fulfilling these tasks during the implementation of an occupational health intervention?	Inductive approach. Focused on role-taking.	The three management groups described each other’s role in a coherent way. Clarifying of roles is necessary before the intervention is implemented. HR managers feel responsible but are little involved in the implementation. All three groups expressed disappointment with how the other actors fulfilled their roles. Managers seldom performed the described roles in practice even they reported high interest toward the intervention. Clear role descriptions and implementation strategies, and aligning an intervention to organizational processes, are crucial for efficient intervention management.
5	Havermans, B.M. (2018) Work stress prevention needs of employees and supervisors BMC Public Health	Qualitative study: Semi-structured telephone interviews. Thematic content analysis.	7 employees and *7 supervisors* (focused on in this review) from different sectors, such as the finance, health care, and services industry.	Employee and supervisor needs regarding organizational work stress prevention. Main issues: (1) communication on work stress, (2) attention to the determinants of work stress, (3) supporting circumstances for the prevention of work stress, (4) involvement of stakeholders in the prevention of work stress and (5) availability of work stress prevention measures.	Inductive approach. Focused on conditions that managers need to prevent work stress.	Supervisors need: Organizational measures with regard on psychosocial work factors (e.g., social support and autonomy). Improvement of the cooperation and the working atmosphere in the team. A safe setting in which to talk about work stress in a team without fear of negative consequences (Communication facilitate awareness and selection of stress management interventions). Support and participation of senior management and other stakeholder in the dialogue on work stress.
6	Horstmann, D. & Remdisch, S. (2019) Drivers and barriers in the practice of health-specific leadership: A qualitative study in healthcare Work	Qualitative study: semi-structured interviews, analyzed by qualitative content analysis.	Interviews with 51 managers from 18 geriatric-care facilities in Germany.	Managers’ perceptions of drivers and barriers in the successful practice of health-specific leadership in the healthcare sector. General understanding of leaders’ influence on employee health. Successful practice and drivers for health- specific leadership.	Drivers and barriers are identified at the leader level, the employee level, and the organizational level. The factors identified relate to the theoretical aspects of the health-specific leadership model: health value, health awareness, health behavior, and role modeling.	Perceived drivers on three levels: Leader level: for all 4 health-specific leadership aspects: Economical perspective, meaning of work, positive vision Healthcare-specific knowledge Personal distance, serenity, stress regulation Willingness to take risks, pragmatism, critical self-reflection, flexibility, decisiveness, persistence, creativity and innovative capacity, exchange within external networks Staff level: Self-responsibility Responsibility, Readiness for change Organizational level: Sustaining chief manager Personal relationships Adequate resources in terms of finances/time/personnel, stability to plan, opportunities for defining work flexibly, good team atmosphere, multipliers in the team, dialogue with the management team, openness and employee involvement
7	Kalef, L. et al. (2016) Employers’ Perspectives on the Canadian National Standard for Psychological Health and Safety in the Workplace Employee Responsibilities and Rights Journal	Qualitative study: through in-person and telephone, explorative semi-structured interviews	10 Toronto and Montreal area business employers of varying workplace sizes.	Canadian employers’ perspectives on the Canadian National Standard for Psychological Health and Safety in the Workplace. (1) employers’ existing understanding of the Standard; (2) the difficulties and advantages of implementing the Standard in their workplace; (3) if the Standard was useful for the employers or not; and (4) the effects of the Standard on the workplace to date.	Concept of Diffusion of Innovation (DOI, Rogers, 2003). A framework that explains how new “innovations” or processes spread throughout social systems such as the workplace.	Employers clearly consider the standard to be beneficial to both workers and companies and consistent with existing initiatives to promote mental health in the workplace. The limited trialability of the standard, the complexity of its introduction in the workplace and the lack of clarity about how visible the results of the introduction of the standard will be may impact the speed of implementation. Employers recognized that a corporate culture that valued mental health and safety would enable progress.
8	Kunyk, D. et al. (2016) Employers’ perceptions and attitudes toward the Canadian national standard on psychological health and safety in the workplace: A qualitative study International Journal of Law and Psychiatry	Qualitative, exploratory study: series of 5 focus groups	17 managers from the fields of healthcare, construction/utilities, manufacturing industries, business services, and finance of a large Western Canadian city.	Employers’ receptivity to implementing the Canadian national standard on psychological health and safety in the workplace.	Inductive approach. Focused on Factors influencing Workplace Mental Health, Reaction to the Standard, Benefits and Barriers to Standard Implementation, Facilitators and Suggestions for the Implementation.	Many employers recognize that mental health and safety in the workplace is a critical issue that needs to be addressed and are looking for guidance on how to address it. The mental health and safety standard is in line with their company’s values and beliefs and can provide guidance. The scope and complexity of the standard can be an obstacle. A simplified implementation process could help to increase the acceptance of the standard, making it a better fit for different organizational cultures and sizes. It was agreed that leadership from the highest level of the organization is critical for the Standard to be adopted well.
9	Landstad, B.J. et al. (2017) How managers of small-scale enterprises (SSEs) can create a health promoting corporate culture International Journal of Workplace Health Management	Qualitative study: analyzed by using an inductive strategy, in accordance with the proposed concepts grounded theory (Glaser & Strauss, 1967) and step-deductive induction (Tjora, 2012).	8 managers from Norwegian and 10 managers from Swedish small-scale enterprises with less than 20 employees. Branches: building and construction/industry; service delivery	Perspective from managers in small-scale enterprises towards workplace health management (WHM) What are prerequisites to WHM? What are possibilities and obstacles for WHM?	Inductive approach. Focused on conditions for workplace health management.	SSE managers are willing to create a good working environment. SSE managers foster antecedent factors and use varied strategies and relationship-based practices as they seek to create a health-promoting culture. Managers highlight difficulties and barriers associated with financial limits, the work environment, and rehabilitation statutes, as well as the demands placed on them to accomplish many tasks while alone in a leadership position.
10	Larsson, R. et al. (2016) Managing workplace health promotion in municipal organizations: The perspective of senior managers. Work	Qualitative study: Semi-structured interviews were conducted individually using open-ended questions based on an interview guide	14 senior managers (part of the upper management) of two Swedish municipal organizations (Stockholm region) from different departments: childcare and education, elderly and social care, traffic and urban planning, environment, human resources (HR), and municipal district administration	How is workplace health promotion (WHP) managed and put into practice by senior management in a municipal organizational context? WHP including work environment: description of WHP and its organization, relations to general organizational policies, and needed changes to WHP and OHS. Leadership strategies: views on leadership development within the organization (e.g., training and important skills).	Inductive approach. Focused on five issues: The dominance on fitness programs, following a problem-solving cycle, building leadership competence, providing managerial support, organizational politics.	Health-related organizational interventions receive less attention than those that focus on individual health behavior. Senior Managers (SMs) followed a problem-solving cycle, whereby an annual employee survey was used to map working conditions and employee health, and the survey served as an important managerial control tool. Senior managers noted multiple difficulties associated with creating and implementing WHP action plans. One difficulty is the centralization of the staff interview process: there is little time to implement all WHP measures before the next annual staff interview. Managers need organizational support to better monitor WHP measures implemented.
11	Moore, A. et al. (2010) Managers’ understanding of workplace health promotion within small and medium-sized enterprises: A phenomenological study Health Education Journal	Qualitative study: a Heideggerian interpretive phenomenological methodology, in-depth telephone interviews	18 managers from small and medium-sized enterprises of a Health and Social Care Trust area of Northern Ireland	Managers’ views on workplace health promotion (WHP) and their experiences with WHP.	Inductive approach. The “Social Diagnosis” of workplace health, adapted from Green & Kreuter (1991) is modified to include an ecological consideration of workplace health determinants, at employee, environmental, business and community levels.	Managers consider WHP to be an important instrument for realizing the potential of both their company and their employees. There is a close relationship between employees and their company. Managers believe that employees’ health is affected by their work as much as their individual health is affected by their ability to work. Managers are more likely to see WHP as using the potential of healthy and safe employees to effectively increase the health and prosperity of their company. They are less concerned with controlling employee health through regulations and constrained practices for the sole purpose of corporate profit.
12	Pescud, M. et al. (2015) Employers’ views on the promotion of workplace health and wellbeing: a qualitative study BMC Public Health	Qualitative study: Phenomenological approach; 10 focus groups	79 employers from a range of industries and geographical locations within Western Australia.	Employers’ perceptions of promoting health and well-being in the workplace and the drivers of those perceptions.	Inductive approach. Focused on three main factors influencing employers’ views on health promotion in the workplace: (1) employers’ conceptualization of workplace health and wellbeing, (2) employers’ descriptions of (un)healthy workers and perceptions surrounding importance of healthy workers, (3) employers’ beliefs around the role the workplace should play in influencing employee’s health.	For many employers (especially in rural areas), the issue of occupational health seems to be embedded in a health and safety paradigm. The issue also appears to be more prevalent in larger workplaces. Women have a more holistic understanding of workplace health and wellness than men. Employers, while aware of the benefits of healthy workers, are unsure whether they have a personal or corporate responsibility to provide health-promoting interventions to their employees. Employers from smaller workplaces were more likely to describe feeling personally responsible for their employees’ health, particularly their mental health (because of friendship). This is in contrast to employers from larger workplaces who consider it less appropriate to make lifestyle suggestions to their employees. Employers were more willing to consider implementing health promotion if they believed it would improve the health or morale of their employees and if the company could afford the cost of implementation.
13	Quirk, H. et al. (2018) Barriers and facilitators to implementing workplace health and wellbeing services (HWB) in the NHS from the perspective of senior leaders and wellbeing practitioners: A qualitative study BMC Public Health	Qualitative study: semi-structured interviews, analyzed by thematic analysis.	Interviews with 4 senior leaders, 4 heads of department and 3 health and wellbeing practitioners of the National Health Service (NHS) in one region of the UK.	Perspective of NHS managers and health and well-being experts about obstacles and enablers to implementing HWB for NHS employees.	Inductive approach. Cultural approach for designing and implementing HWB regarding facilitators and barriers on different levels: individual, interpersonal, organizational, cultural, policy.	Described obstacles to implementation: hectic pace and pressure due to staff shortages; financial obstacles to implementing HWB; awareness of priorities: patients before staff. Perceived obstacles to employee engagement: logistical obstacles at NHS; employees need to be open-minded. Helpful factors for introducing HWB services in the NHS: government programs and funding as incentives; an organizational infrastructure that fosters HWB; an organizational culture that encourages HWB among employees; a coherent, strategic approach to implementation; corporate communication; creative and innovative management of resources; needs assessment, and review.
14	Rodham, K. & Bell, J. (2002) Work stress: an exploratory study of the practices and perceptions of female junior healthcare managers Journal of Nursing Management	A combination of critical incident diaries and semi-structured interviews. The themes emerging from the diary entries were identified using a grounded theory approach.	Sample of 6 junior managers (JM) from a large NHS hospital in London. Nonclinical manager (N = 2), clinical manager (N = 4).	Investigation of the beliefs and behaviors of junior managers in the health care sector towards stress in the workplace.	Inductive approach. Focused on attitudes and beliefs concerning the definitions of stress, recognition, responsibility for management, personal stressors and awareness of stress.	There is a shortage of consciousness about stress in the workplace among junior medical (JM) leaders. JM concentrate their stress management on assisting employees with signs of stress, rather than on the underlying stressors (reasons for stress). JM do not see workplace stress management as the duty of the company. JM associate stress with individual aspects and starting points rather than organizational ones. JM notice a climate of embracing and taking work stress for granted, coupled with a deficit of understanding of how to deal with stress efficiently and actively.
15	Schulte, M. & Bamberg, E. (2002) Ansatzpunkte und Nutzen betrieblicher Gesundheitsförderung aus der Sicht von Führungskräften. Gruppendynamik und Organisationsberatung	Qualitative study: semi-structured individual interviews; content-analysis according to Miles & Huberman (1994)	40 senior and top-level managers (director and board level) of a Scandinavian and a German aviation company	Manager’s perspective on: the occupational health situation the responsibility for and the effectiveness of health promotion measures factors hindering and promoting the implementation of health promotion measures/company policy in the company existing intentions, on the one hand to develop and initiate health-promoting measures, on the other hand to support and implement a health-promoting corporate policy? How are the intentions concretized?	Schwarzer’s Health Action Process Approach HAPA (1996), with the following health psychological approaches: the “Health Belief Model” (Rosenstock, 1966; Becker 1974; Janz & Becker, 1984), the “Theory of Reasoned Action” (Fishbein & Ajzen, 1975; Fishbein & Ajzen, 1980; Ajzen, 1985) and the “Protection Motivation Theory” by Rogers (1975; 1983; 1985)	Managers do not experience occupational health as a threatening (existential) issue, but still wish for improvements in psychosocial health. They see threatening proportions more in economic developments. Managers associate a “healthy company” primarily with economic efficiency (more German than Scandinavian managers). Profitability is seen as the basis for the well-being of employees, not the opposite. Almost half of the managers would like to see an improvement in the psychosocial health situation. Managers consider soft factors (individual orientated) of health promotion (human interaction, open communication and role model function) to be more effective than interventions that could contribute to broader health promotion. A health-promoting corporate policy is not considered to be effective by the majority of respondents. Scandinavian managers refer more to comprehensive measures of management development and organizational development in their planning projects, German managers refer mainly to individual measures. Scandinavian managers have a consistently positive attitude towards the feasibility of extended health measures. The predominant health culture in the company (e.g., in the form of already successfully institutionalized measures) seems to have an influence on the anticipated feasibility of further psychosocial health promoting measures
16	van Berkel, J. et al. (2014) Ethical considerations of worksite health promotion: an exploration of stakeholders’ views BMC Public Health	Semi-structured focus group discussions: Data analyzing according to the constant comparison method.	Employees (N = 6) and Employers (N = 4) from large and smaller organizations (industry and service) involved in WHP.	A comparison of stakeholder views on workplace health promotion and resulting ethical aspects. Themes of the analysis: the definition of occupational health, occupational health risk factors, worksite health promotion, and taking responsibility.	Inductive approach: focused on ethical aspects of worksite health promotion.	Views on risk factors for occupational health vary between stakeholders: Workers and trade union representatives see risk factors as mainly workplace-related, while employers see employee-related risk factors (e.g., lifestyle behaviors). All stakeholders (incl. employers) generally consider the responsibility of the employer to provide a healthy working environment, as they are required by law. There is consensus that employees are responsible for their healthy behavior, but there is a different understanding of responsibility. For employees, responsibility means autonomy, while for employers and other stakeholders, responsibility is synonymous with duty.
17	Zhang, Y. et al. (2016) Workplace participatory occupational health/health promotion program: Facilitators and barriers observed in three nursing homes Journal of Gerontological Nursing	Qualitative study: focus group discussions (employees) and in-depth interviews (manager) as part of the evaluation of the participatory occupational health/health promotion program.	In-depth interviews with 5 top managers (i.e., administrators and directors of nursing and 13 middle managers (i.e., department heads and unit managers) of three nursing homes in the eastern united states.	Perception of the facilitators and barriers for the participatory occupational health/health promotion program from managers’ and employees’ perspectives.	The Social Ecological Model (SEM) for health promotion (McLeroy et al., 1988). Regard of different levels: Intrapersonal, interpersonal, organizational, corporate.	Three most essential factors for successful implementation: management support, adequate finances, and time resources to participate in the program. Additional important aspects: A working board with an engaged coordinator is essential for good workplace communication and motivating staff to take part; Support at multiple organizational levels, driven by human and environmental factors.

**Table 2 ijerph-19-12610-t002:** Theoretical approaches and their relation to the TPB-based framework.

Components of the TPB-Based Framework (in Genrich et al., 2020)	Health Specific Leadership Model ^1^ (in Horstmann & Remdisch, 2019) and Health-Oriented Leadership ^2^ (in Efimov et al., 2020)	Social Ecological Model for Health Promotion (in Zhang et al., 2016)	The Health Action Process Approach (in Schulte & Bamberg, 2002)	Diffusion of Innovation (in Kalef et al., 2016)	Inductive Approaches (in Included Studies of the Other Authors)
Attitude—Belief in importance	Health value ^1,2^ Health awareness ^1,2^		Perception of a threatening situation for their own health		
Attitude—Belief in outcome	Health value ^1,2^			Relative advantage	
Attitude—Belief in role	Role modeling ^1^				Role taking, Managers’ responsibility
Organizational norms				Compatibility	
Perceived behavioral control (PBC) due to internal or organizational facilitators or barriers	Drivers and Barriers at leader, employee and organizational levels in relation to the components of the model (see above), incl. health behavior. ^1^ Decisive factors for the implementation of HoL ^2^	Facilitators and Barriers with consideration of different levels of influence, including intrapersonal, interpersonal, organizational, corporate.	Barriers Feeling of self-efficacy, Knowledge about measures, Facilitators	Observability Trialability Complexity	Critical individual or organizational conditions

^1^ Health Specific Leadership Model; ^2^ Health-Oriented Leadership. Explanation: The assignment of the components is done via the numbers to the two models.

## Data Availability

Not applicable.

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
