# Peer review of "Managers’ Action-Guiding Mental Models towards Mental Health-Related Organizational Interventions—A Systematic Review of Qualitative Studies"

_ijerph, 2022, doi:10.3390/ijerph191912610_

Round 1

Reviewer 1 Report

Greetings!

After reviewing this manuscript, I can present the following comments:

1- The abstract should include purpose of study, data collection method, main findings, and   recommendations. The abstract does not include clear results and recommendations. The authors should provide clear results and recommendations in the abstract.

2- Given the importance of the study, I believe that 17 literatures in this field are insufficient to judge and rely on in formulating the results of this study (You depend on very big countries in preparing your study such as USA, Australia, and several Europe countries). The authors should work on increasing the number of these previous studies as much as possible so that they are more representative as a sample for this study. Remember this is a review study.

3- Are there no studies on the subject after 2020?

4- The most recent references used are from 2020 and before. Are there no references on the subject after 2020?

All the best,

Reviewer 2 Report

Thank you for the opportunity to review this article.  Your subject matter is very interesting and timely.  Given the ongoing challenges highlighted by covid-19 on employment retention and employee wellness, this study is well received.  Additionally, the focus on management views makes a needed move from the focus of wellness actions on employees to those bridging the middle between the individual and the agency.  

  • Good overview of theory and why selected as well as application to the study itself.

  • "Subject beliefs" are well evidenced and applicable to this topic.

  • Good review of inclusion and exclusion criteria- reasoning is clearly stated.

  • This is a well-written and clearly stated study.  Objectives, terms, theory and methodology are clearly described and evidence provided is sufficient and reflective of current and past research

  • Your proposed implementation in practice is well thought and reflective of your data analysis.  You provide clear ideas of how employee health can be addressed from a top-down perspective and address the co occurrence of employee wellness and company productivity, guiding away from the polarization of these topics.

Very well written and well researched study. I recommend for publicaton.
